# Towards a Sustainable Management of the Spotted-Wing Drosophila: Disclosing the Effects of Two Spider Venom Peptides on *Drosophila suzukii*

**DOI:** 10.3390/insects14060533

**Published:** 2023-06-07

**Authors:** Laura Regalado, Sara Sario, Rafael J. Mendes, Javier Valle, Peta J. Harvey, Cátia Teixeira, Paula Gomes, David Andreu, Conceição Santos

**Affiliations:** 1iB2, Biology Department, Faculty of Sciences, University of Porto, 4169-007 Porto, Portugal; laura.regalado@fc.up.pt (L.R.); rafael.mendes@fc.up.pt (R.J.M.); csantos@fc.up.pt (C.S.); 2LAQV-REQUIMTE, Faculty of Sciences, University of Porto, 4050-453 Porto, Portugal; ca.teixeira@gmail.com (C.T.); pgomes@fc.up.pt (P.G.); 3Proteomics and Protein Chemistry Unit, Department of Medicine and Life Sciences, Pompeu Fabra University, 08002 Barcelona, Spain; javier.valle@upf.edu (J.V.); david.andreu@upf.edu (D.A.); 4Institute for Molecular Bioscience, Australian Research Council Centre of Excellence for Innovations in Peptide and Protein Science, The University of Queensland, Brisbane, QLD 4072, Australia; peta.harvey@imb.uq.edu.au

**Keywords:** gene expression, J-atracotoxin-Hv1c, neurotoxic peptides, spotted-wing drosophila, stress-related pathways, µ-theraphotoxin-Hhn2b

## Abstract

**Simple Summary:**

*Drosophila suzukii* is a major destructive insect pest with a pandemic distribution. The lack of effective green control measures for this pest has prompted the search for new approaches, among which are peptides from animal venom. In this study, the biological activity of two underexplored spider venom peptides (J-atracotoxin-Hv1c and µ-theraphotoxin-Hhn2b) was assessed against adult *D. suzukii* flies, as well as the biological response of flies to these peptides through detoxification mechanisms. Results demonstrate that µ-theraphotoxin-Hhn2b enhanced fly longevity. Gene expression analysis suggests that detoxification and stress-related mechanisms are triggered in *D. suzukii* flies in response to treatment with these peptides. Our results highlight the potential of venom peptides to control *D. suzukii*, underscoring the issue of how to ultimately devise improved target-specific formulations.

**Abstract:**

The spotted-wing drosophila (*Drosophila suzukii*) is a polyphagous pest that causes severe damage and economic losses to soft-skinned fruit production. Current control methods are dominated by inefficient cultural practices and broad-spectrum insecticides that, in addition to having toxic effects on non-target organisms, are becoming less effective due to acquired resistance. The increasing awareness of the real impact of insecticides on health and the environment has promoted the exploration of new insecticidal compounds, addressing novel molecular targets. This study explores the efficacy of two orally delivered spider venom peptides (SVPs), J-atracotoxin-Hv1c (Hv1c) and µ-theraphotoxin-Hhn2b (TRTX), to manage *D. suzukii*, through survival assays and the evaluation of gene expression associated with detoxification pathways. Treatment with TRTX at 111.5 µM for 48 h enhanced fly longevity compared with the control group. Gene expression analysis suggests that detoxification and stress-related mechanisms, such as expression of P450 proteins and apoptotic stimuli signaling, are triggered in *D. suzukii* flies in response to these treatments. Our results highlight the potential interest of SVPs to control this pest, shedding light on how to ultimately develop improved target-specific formulations.

## 1. Introduction

*Drosophila suzukii* Matsumura (Diptera: Drosophilidae), also known as spotted-wing drosophila (SWD), is one of the most relevant invasive pest species in the world [1]. SWD was first described in 1931 in Japan [2], and in 2008 was observed in the USA, Spain, and Italy [3], quickly reaching a worldwide distribution during the last decade [4,5]. SWD is a polyphagous species and has different small fruits as preferred hosts for its development, such as blueberry (*Vaccinium* spp.), raspberry (*Rubus* spp.), blackberry (*Rubus* spp.), and strawberry (*Fragaria* spp.) [2,6].

This pest takes advantage of the female’s highly sclerotized ovipositor to lay eggs inside healthy soft-skinned fruits, unlike other drosophilids such as the closely related non-pest *Drosophila melanogaster* [1]. Inside the penetrated fruit, the eggs hatch and develop into larvae, which will feed on the fruit core, making it unconsumable and unmarketable [7,8]. Furthermore, the wounds created through oviposition act as an entry point for invasion by other insects and also by opportunistic bacterial and fungal infections, further aggravating production losses since these diseases can affect entire orchards if not controlled [1,9].

Current control of *D. suzukii* primarily relies on cultural practices (e.g., mulching and pruning) and on the application of broad-spectrum synthetic insecticides, including organochlorines, organophosphates, carbamates, and pyrethroids, most of them facing increased restrictions in the EU, due to their adverse environmental effects [10,11,12]. Moreover, the application of these products can also have negative impacts on both mammals and beneficial arthropods, such as crucial pollinators. Available reports point out the existence of resistance-acquired problems in *D. suzukii*, ultimately imposing limitations on the efficiency of these treatments [13,14].

The increasing awareness of the environmental and health impacts of insecticides has encouraged the search and study of new effective pest control molecules aimed at novel molecular targets. In this context, spider venom peptides (SVPs) emerge as potential new naturally occurring green compounds to manage insect pests, such as *D. suzukii* [15,16,17,18,19,20,21]. These peptides are produced by the venom glands of spiders, playing an important role in the defense against predators and prey hunting [22,23]. SVPs are complex peptides stabilized by multiple disulfide bridges. Their backbone creates a compact motif that provides exceptional chemical, thermal, and enzymatic stability [24]. In recent years, several studies on venom peptides have explored their bioactivity alone or in conjugation with adjuvants or carrier molecules [15,16,17,18,19,20,21]. Most of these peptides modulate the activity of neuronal receptors or ion channels, either resulting in decreased stimulation of the nervous system, causing paralysis, or surcharging it, leading to convulsive paralysis and death [25]. However, the potential of SVPs for controlling *D. suzukii* has not received much attention, with only one study to date exploring the activity of GS-omega/kappa-Hxtx-Hv1a (Hv1a) in SWD [21].

To fill this gap, this study explored the application of two SVPs with unknown effects on *D. suzukii*, namely J-atracotoxin-Hv1c (Hv1c) and µ-theraphotoxin-Hhn2b (TRTX, formerly hainantoxin I or HNTX-I). Hv1c, a 37-residue peptide with four disulfide bridges, was first isolated from the Australian web-tunnel spider *Hadronyche versuta* and has already proven to be efficient against other Diptera, such as *Musca domestica* and *D. melanogaster* [26,27]. TRTX is derived from the venom of the Chinese bird spider *Selenocosmia hainana* and has 33 amino acid residues stabilized by three disulfide bonds. In a previous study, TRTX displayed 15-fold higher selectivity to *Drosophila* para channels over rat voltage-gated sodium channels (VGSCs) ([28], as interpreted by Windley [29]), hinting at a possibly safer effect on mammals than on insects.

Taking into account the worldwide concern with the application of the current insecticides, and in line with the European Green Deal, with particular reference to the Farm to Fork strategy to foster more sustainable agriculture, this work aims to disclose the potential of Hv1c and TRTX to control *D. suzukii*. To achieve this goal, the two peptides were first produced in adequate amounts by efficient chemical synthesis, then the survival and longevity of flies exposed to peptides were assessed. Additionally, to evaluate the influence of treatments on *D. suzukii* key cell pathways associated with stress resistance and xenobiotic detoxification, gene expression analysis was conducted by real-time quantitative PCR (RT-qPCR). The data collected here may elucidate the potential of natural peptides as green control strategies for *D. suzukii* while emphasizing their promising role in insect pest management in the agro-food industry.

## 2. Materials and Methods

### 2.1. Biological Material

A *D. suzukii* colony was established from a blueberry orchard (*Vaccinium* spp.) located in São Martinho de Mouros, Resende, Portugal (41°07′13.4″ N 7°53′17.8″ W). Eggs and larvae were obtained by collecting ripe-infested blueberries from this field. The colony was housed in plastic vials containing standard drosophila cornmeal diet [1.5% (*w*/*v*) agar, 4% (*w*/*v*) brewer’s yeast, 8% (*w*/*v*) cornmeal, 10% (*w*/*v*) sugar, 0.08% (*w*/*v*) methyl 4-hydroxybenzoate, 0.15% (*v*/*v*) orthophosphoric acid, and distilled water (dH_2_O)] [30], at 23 °C under a 16:8 h of light:dark photoperiod, and was transferred to new vials every week. In every survival experiment, 2–5 days-old flies were used (five males and five females per experimental vial).

### 2.2. Chemicals

*N^α^*-Fmoc-protected amino acids (Fmoc-L-AA-OH), *N*,*N*,*N′*,*N′*-Tetramethyl-*O*-(1*H*-benzotriazol-1-yl)uronium hexafluorophosphate (HBTU), 2-chlorotrityl chloride (2-CTC) resin, and H-L-Pro-2-chlorotrityl resin were acquired from Irish Biotech GmbH (Marktredwitz, Germany). Gradient grade acetonitrile (ACN) and peptide synthesis-grade dimethylformamide (DMF), dichloromethane (DCM), *N*,*N*-diisopropylethylamine (DIEA), and trifluoroacetic acid (TFA) were purchased from Carlo Erba-SDS (Sabadell, Spain). Other chemicals, such as triisopropylsilane (TIS), 3,6-dioxa-1,8-octanedithiol (DODT), guanidinium hydrochloride (Gd·HCl), among others, were purchased from Sigma–Aldrich (Madrid, Spain), unless otherwise indicated.

### 2.3. Peptide Synthesis

The precursors of Hv1c, AICTGADRPCAACCPCCPGTSCKAESNGVSYCRKDEP (octathiol), and TRTX, ECKGFGKSCVPGKNECCSGYACNSRDKWCKVLL (hexathiol), were assembled on a Prelude (Gyros Protein Technologies, Tucson, AZ) instrument running Fmoc SPPS protocols, using a 2-CTC resin (100–200 mesh, 0.6 mmol/g) preconditioned in DCM and DMF. The first *C*-terminal amino acid (AA) was coupled manually to the resin in 8-fold molar excess before automated synthesis, using a mixture of the Fmoc-L-AA-OH (eight molar equivalents, eq) dissolved in DMF, and base (DIEA, 16 eq). Automated double coupling of the subsequent residues was performed with a mixture of Fmoc-L-AA-OH and HBTU (8 eq each) in the presence of DIEA (16 eq) in DMF as solvent. Fmoc protecting groups were removed with piperidine/DMF (20:80 *v*/*v*) followed by DMF washes.

Once the target sequences were completed, full deprotection on the side chains with concomitant cleavage of the peptide from the resin was done by acidolysis with TFA/H_2_O/DODT/TIS (94:2.5:2.5:1 *v*/*v*) at room temperature (RT) for 2 h under constant shaking (1 mL of cleavage cocktail/100 mg resin). The crude peptide was isolated from the TFA solution by precipitation with chilled diethyl ether and centrifugation at 4700 rpm for 10 min at 4 °C. This procedure was performed thrice. The peptide pellet was taken up with water and lyophilized.

Peptide purification was done by preparative reversed-phase high-performance liquid chromatography (RP-HPLC) on a reverse-phase C18 column (250 × 21.2 mm, 5 μm pore size, Phenomenex^®^) in a Shimadzu (Kyoto, Japan) instrument equipped with an SPD-20A UV/Vis detector and an LC-20AP pump. A linear gradient from 15 to 50% of solvent B into A (A: 0.1% TFA in H_2_O; B: 0.1% TFA in ACN) over 30 min was applied at 20 mL/min flow rate, with detection at 220 nm.

Peptide identity was confirmed by liquid chromatography hyphenated with mass spectrometry (LC-MS) and matrix-assisted laser desorption ionization time of flight (MALDI-TOF) MS.

### 2.4. Peptide Oxidative Folding and Purification

The linear peptides were diluted to 10^−5^ M in water and oxidized in a solution containing 0.1 M ammonium acetate (NH_4_OAc) with 1 M Gd·HCl in the presence of reduced (GSH) and oxidized glutathione (GSSG) (1:100:10 molar ratio of peptide/GSH/GSSG). The pH was adjusted to 7.8–8.0, and the oxidation was performed at RT for 20–24 h under an anaerobic environment (N_2_ bubbling) and shuffling. The oxidation was monitored by analytical HPLC (15–50% and 5–50% linear gradient of solvent B into A over 15 min, at a 1 mL/min flow rate for Hv1c and TRTX, respectively) and LC-MS (gradient: 5–50% of solvent B into A over 15 min, at a 1 mL/min flow rate). Acetic acid was used to quench the oxidation when the process was judged to be completed.

To retrieve the folded peptides from the oxidation reaction, the solutions were applied through a peristaltic pump model 313S (Watson Marlow) to a Strata^®^ reverse-phase C18-E cartridge (Phenomenex^®^). The cartridge was washed with 2 mL of 0.1% TFA in ACN and 2 mL of 0.1% TFA in water. The peptide solutions were then pushed through the column in a flow of 4 mL/min to retain the peptide, and then the peptides were eluted with ACN/H_2_O with 0.1% TFA (49.5:49.5:1 *v*/*v*) and lyophilized. Lyophilized peptides were dissolved in Milli-Q water and purified by RP-HPLC as previously done for their linear precursors. Final products were stored at −20 °C until further use.

The molecular weight of both SVPs in their linear and oxidized forms was confirmed by MALDI-TOF MS analysis. To this end, spectra were acquired in a 4800 Proteomics Analyzer (AB Sciex, Darmstadt, Germany) operated in positive ion mode at an acceleration voltage of 20 kV, 80% grid voltage, 1.227 ns delay time, and 2.19 kV detector voltage. Equal volumes of peptide (1 mg/mL) and matrix solution (α-cyano-4-hydroxy-cinnamic acid, 15 mg/mL in 50% MeCN in H_2_O) were mixed on the MALDI plate and air-dried. Spectra were recorded in reflector TOF mode in the 1000–4000 m/z range by accumulating 30 subspectra at a fixed laser intensity of 4900. The spatial arrangement of peptides and the connectivity of disulfide bridges were determined by nuclear magnetic resonance (NMR) (Appendix A).

Peptides were quantified using a NanoDrop™ One UV-Vis spectrophotometer (Thermo Fisher Scientific, Waltham, MA, USA). For Hv1c, quantification method 31 was used, assuming an extinction coefficient ε_205_ of 31 mL mg^−1^ cm^−1^, whereas peptide TRTX was quantified by the Scopes method [31].

### 2.5. Survival Assays

*D. suzukii* flies of both sexes were submitted to treatments with SVPs through oral ingestion. For this purpose, filter paper covering the tube bottom was embedded with a solution of 0.1 M sucrose and yeast extract at 5% (1:1) with equal volumes of each SVP separately, to final concentrations of 2.23, 111.5, and 223 µM. These concentrations were chosen taking into account the study from Maggio and King [27]. dH_2_O with the nutritious solution was used as a negative control. For the positive control, the commercial product Spintor^®^ (Corteva Agriscience, Indianapolis, IN, USA) was used at a final concentration of 5 mL/100 L. *D. suzukii* flies were starved for 30 min in empty vials to provide moderate, stress-reduced feeding motivation and then transferred to the prepared vials for the appropriate assay duration (2, 4, 24, and 48 h), after which flies were transferred to a peptide-free medium to assess SVP effects on their lifespan. Adults were removed and housed in new vials as pupae appeared. Flies’ survival was monitored and recorded daily until 40 days post-SVP feeding. The survival experiment was performed with three biological replicates comprising three technical replicates of 10 flies each (five males and five females), for a total of 90 flies per condition.

### 2.6. Gene Regulation of Key Pathways in Response to SVPs

#### 2.6.1. Total RNA Isolation and cDNA Synthesis

The total RNA of flies was isolated using a ready-to-use Tri-reagent (NZYol, NZYTech, Lisbon, Portugal) following the manufacturer’s instructions with minor modifications. A pool of one male and one female fly was randomly chosen from each replicate of the following conditions: Hv1c and TRTX at 111.5 µM for 48 h (corresponding to the most promising results of the survival assay) and the corresponding negative control. Briefly, samples were homogenized with 800 µL of NZYol with a 2.8 mm ceramic bead in a Fisherbrand^TM^ Bead Mill 24 (Thermo Fisher Scientific, Waltham, MA, USA) at 2.4 m/s for 60 s and then incubated for 5 min at RT. After adding chloroform (Analysis grade, Merck, Germany), samples were incubated for 3 min at RT and centrifuged at 12,000× *g* for 15 min at 4 °C (Mikro 200R, Hettich, North America, Beverly, MA, USA). The aqueous phases were transferred to new microtubes, and DNase I was applied to all samples according to the manufacturer’s instructions (NZYTech, Lisbon, Portugal). RNA was precipitated with cold isopropyl alcohol (Fischer Scientific) for 1 h at −20 °C, and samples were centrifuged at 12,000× *g* for 10 min at 4 °C. Pellets were washed with ethanol 70% and samples were centrifuged at 12,000× *g* for 5 min at 4 °C. The pellets were air-dried and resuspended in 80 µL of RNase-free water.

cDNA was synthesized using an NZY First-Strand cDNA Synthesis Kit (NZYTech, Lisbon, Portugal) following the manufacturer’s protocol. RNA and cDNA quality and concentration were evaluated with a FLUOstar Omega Microplate Reader (BMG LABTECH, Ortenberg, Germany). Working solutions of 60 ng/µL cDNA were made and stored at −20 °C until further use.

#### 2.6.2. Design of Primers

Primer sequences for genes associated with cytochrome P450 pathway (adenylyl cyclase 13E, *ac13E*; cytochrome P450 12d1 distal, *cyp12d1-d*), synthesis of heat shock protein 70 cognate 4 (*hsc70-4*), insulin-like receptor/target of rapamycin (InR/TOR) pathways (substrate of the product of InR, *chico*; forkhead box, sub-group O, *foxo*; superoxide dismutase 2, *sod2*) and the apoptotic stimuli pathway (death caspase-1, *dcp-1*; death-associated inhibitor of apoptosis 1, *diap1*) were used for RT-qPCR analysis (Table 1). Sequences of *D. suzukii* genes were obtained from the SpottedWingFlyBase (http://spottedwingflybase.org/ (accessed on 1 June 2022). Unless referenced, primers were designed using Primer3Plus (version: 3.2.6) [32]. The genes elongation factor 1 beta (*ef1a48D*) [33], TATA-binding protein (*tbp*), and arginine kinase (*argK*) [34] were selected as housekeeping reference genes.

#### 2.6.3. Gene Expression Evaluation by Real-Time Quantitative PCR (RT-qPCR)

RT-qPCR amplifications were performed using a CFX Connect™ Real-Time PCR Detection System (Bio-Rad, Hercules, CA, USA). The reaction mixture consisted of 10 µL of NZYSupreme qPCR Green Master Mix (2×), ROX plus (NZYTech, Lisbon, Portugal), 400 nM of each primer, 120 ng of cDNA template, and sterile Milli-Q water up to a final volume of 20 µL. No-template controls were included in each set of reactions. The amplification program was as follows: initial polymerase activation at 95 °C for 2 min, followed by 45 cycles of denaturation for 5 s at 95 °C, and 30 s at 60 °C for the annealing and extension. Melting curves were obtained by 0.5 °C increments in 5 s/cycle from 65 to 95 °C, with continuous fluorescence detection. All RT-qPCR experiments were carried out with three technical replicates. An RDML-LinRegPCR tool was used to calculate primer efficiency [35]. Relative quantification of gene expression was calculated using the standard ΔΔCq method as described by Taylor et al. [36].

### 2.7. Statistical Analysis

Survival curves were calculated and plotted as Kaplan–Meier curves. The Log-rank (Mantel–Cox) test was used to compare survival probabilities among control and treatments, and interpretation of individual *p* values was performed using the Bonferroni correction method. Flies whose death was not associated with treatment (e.g., stuck on media) were censored. Comparisons between treatments and the control for gene expression levels were made with the one-way ANOVA test and corrected with Dunnett’s T3 multiple comparisons test. Multivariate analyses for data correlation were made through a Principal Component Analysis (PCA). Significance was defined as *p* < 0.05. All data were analyzed using GraphPad Prism version 9.0.0. for Windows (GraphPad Software, San Diego, CA, USA).

## 3. Results

### 3.1. Peptide Synthesis and Oxidative Folding

The synthesis of peptides Hv1c and TRTX (linear precursors, i.e., in the reduced form) was performed by solid phase methods using Fmoc/*^t^*Bu chemistry. Oxidative folding of Hv1c and TRTX yielded the oxidized peptide as the main product in 69.2% and 67.6% yield, respectively (Appendix A). The peptides were purified by RP-HPLC high purity (>97%), had the expected mass, and were confirmed to be properly folded, as they adopted the same 3D solution structures as the corresponding wildtypes, based on NMR analysis (Appendix A) [28,37].

### 3.2. Oral Administration of SVPs May Disrupt D. suzukii Longevity

Oral administration of the two SVPs disclosed different susceptibilities of the flies to the peptides, according to concentrations and period of exposure tested (Figure 1; Appendix A). Flies exposed to 2.23 µM Hv1c for 2 h had higher median survival compared to the control (33 days and 20 days, respectively), which was statistically different (*p* < 0.05). For flies exposed to 2.23 µM TRTX for 2 and 4 h, the median survival time was 20 and 26 days (*p* > 0.05), respectively (Appendix A).

Taking into account these results, the subsequent experiments were performed with increased concentrations up to 223 µM for 24 and 48 h. After 24 h of exposure to both SVPs treatments at 111.5 µM, when compared to the control group, no statistically significant differences were observed (*p* = 0.0192) (Appendix A). Nevertheless, the higher mortality observed from day 30 onwards for flies treated with 111.5 µM TRTX for 24 h relative to the control (55.17% and 18.73%, respectively) exhibited statistically significant differences (*p* = 0.0036 after correction for multiple comparisons).

Flies exposed to 111.5 µM of Hv1c and TRTX for 48 h had a median survival time of 9 and 14 days, respectively, compared to 12 days for the control group flies (*p* = 0.0019) (Figure 1). As already pointed out for a lower concentration of Hv1c, flies exposed to 111.5 µM of TRTX for 48 h compared to control unveiled a higher survival (10.05% and 0%, respectively) and longevity (>40 and 30 days, respectively) (*p* = 0.0351 after Bonferroni correction). Doubling the peptide concentrations to 223 µM had no visible effects on the survival and longevity of flies, as there were no significant differences among control and treatments across the whole experiment (Appendix A).

### 3.3. Gene Expression Evaluation in Response to SVPs

Considering the results from the survival curves, a further assessment was carried out to evaluate the expression of reference stress-related genes by RT-qPCR for flies exposed to both SVPs at 111.5 µM for 48 h and the respective control group (Figure 2).

Gene expression analysis revealed that *D. suzukii* flies had an increased transcription level of gene *diap1* following treatment with TRTX, which was statistically different (*p* < 0.05) compared to the control group (Figure 2h). In addition, other genes displayed slight changes. For example, flies treated with TRTX and Hv1c presented an increased expression of *cyp12d1-d* by 1.81-fold, and *dcp-1* by 0.88-fold, respectively, when compared to control (Figure 2b,g), although no significant differences were observed (*p* > 0.05).

### 3.4. SVP Bioactivity Is Hampered by Key Detoxification and Stress-Response Pathways in D. suzukii

PCA was performed to trace a correlation between SVPs and their influence on the expression of stress-related genes. A clear separation between the control and treatment groups was observed (Figure 3). The first axis (PC1), where most of the variance is explained (62.36%), strongly separates treatment Hv1c from TRTX. The second axis (PC2) explains 37.64% of the variance and separates the control from the treatments. Control is located at the upper right quadrant, and gene expression of *ac13E* is strongly correlated with this condition. The Hv1c is located on the lower right quadrant, and although there is not a clear clustering of genes linking to its treatment, it is the condition scoring the highest expression for genes *hsc70-4* and *dcp-1*. Moreover, condition TRTX is located on the lower left quadrant, and gene expression of *cyp12d1-d*, *sod2*, *chico*, and *diap1* genes are more closely associated with it.

## 4. Discussion

*D. suzukii* is one of the most threatening insect pests worldwide. Given the inefficiency of the existing management approaches coupled with their detrimental impact on non-target species and the environment, it is imperative to find new classes of eco-friendly insecticides that achieve a balance between efficiency and biosafety to control insect pests such as SWD. In the field of insect management, diverse animal venom peptides have been studied against several insect species, namely *Lucilia cuprina* [19], *D. melanogaster* [15,19,38], *Acyrthosiphon pisum* [39], and *Periplaneta Americana* [40], among others. However, only one study to date has focused on finding an SVP targeting the species *D. suzukii*. This work from Fanning et al. [21] explored the activity of GS-omega/kappa-Hxtx-Hv1a (Hv1a) in synergy wit—h phagostimulants and adjuvants and described a significant increase of mortality from 10.8% for Hv1a-treated flies to >90% for flies exposed to a combination of Hv1a with two agricultural adjuvants separately. Moreover, most of the studies in this area explore the bioactivity of peptides by directly injecting them into the insect hemolymph or through topical exposure [16,19,20,21,39,40,41,42,43,44]. Hence, the present work aims to overcome existing knowledge gaps by exploring two SVPs whose effects on *D. suzukii* remain undisclosed. In contrast with what has been explored so far, this study describes a method to evaluate the insecticidal activity of a compound through the oral route.

To elucidate the insecticidal activity of Hv1c and TRTX, a series of bioassays with oral administration of these compounds was developed. The survival assays showed a different susceptibility of *D. suzukii* to both SVPs tested. Although treatment with 111.5 µM TRTX for 24 h showed statistically significant differences, in the first 30 days after treatment, the mortality was not affected. This late effect might be attributed to the oral ingestion method, which means that the peptide has to face several physical and chemical barriers before reaching the central nervous system [45], where it is believed to display its main effect as an insecticide. However, from an agricultural practice point of view, it is of most interest to have higher mortality in the days immediately following treatment. An earlier disturbance in the flies’ survival results in fewer flies reproducing and laying their eggs on the fruit, minimizing the offspring and, therefore, the impact on fruit crops.

On the other hand, the increased longevity observed for flies treated at lower concentrations [2.23 µM in the case of Hv1c (4 h); TRTX at 111.5 µM for 48 h] may be explained by the concept of hormesis often observed in dose-effect studies. This phenomenon comprises a biphasic response characterized by low-dose stimulation and a high-dose inhibition of the expected effect [46]. This differential response is advantageous for insects exposed to sublethal doses of a certain compound, resulting in a stimulation of their protective and detoxification mechanisms, which can be reflected in behavioral changes and stimulation of reproductive activity, as examples [47,48]. For instance, Krüger et al. [49] conducted a study on *D. suzukii,* disclosing that a neurotoxic insecticide with action on insect coordination was, however, able to positively affect mating when applied at sublethal doses. Similarly, a study developed by Deans and Hutchison [50] on SWD revealed evidence of the hormesis response to sublethal doses of the insecticides zeta-cypermethrin, spinetoram, and pyrethrin. This study also described survival as being sex-dependent, as males were more susceptible to insecticides than females. Treatment with phenobarbital and atrazine, medicinal products for human purposes and a herbicide, respectively, promoted sex-related differential transcription of genes encoding for P450 and glutathione S-transferase enzymes in *D. melanogaster* [51]. Moreover, this study described a selective expression of P450-encoding genes according to the different xenobiotics as a result of adaptative response to chemically adverse environments [51].

The evaluation of the transcription level of some stress-related genes, conducted for flies treated with SVPs at 111.5 µM for 48 h, revealed a clear peptide-dependent response (Figure 1). These results showed an up-regulation of *cyp12d1-d* for TRTX-treated flies. Although this change was not statistically significant, it is hypothesized that this may be linked to peptide detoxification as a biological response from *D. suzukii* to a foreign compound (Figure 4), thus promoting the higher survival and longevity registered for flies of this condition (Figure 1). In fact, some studies correlate the expression of detoxification genes with increased insecticide tolerance, metabolic resistance, and stimulation of longevity and fertility [52,53,54]. This resistance is linked to increased metabolic detoxification and accelerated elimination of xenobiotics [55]. For instance, Civolani et al. [56] described that SWD flies submitted to increasing concentrations of cyantraniliprole had a dose-dependent response in the expression of the gene *cyp12d1*, encoding for a P450 protein. This study also held an insecticide resistance selection assay with a cyantraniliprole-susceptible colony as a starting point, and by the eighth generation, the LC_50_ value increased 2.2-fold compared with the control population, suggesting that this adaptive response may be related to the expression of detoxifying agents such as P450 enzymes [56].

In *Drosophila*, InR/TOR routes interpret growth factors, oxygenation, nutritional levels, and hormones, ultimately leading to changes in cell growth, proliferation, and control of egg development [57]. Under adverse conditions, the expression of downstream oxidative stress-related genes, such as those encoding for superoxide dismutases, is triggered in response to the disinhibition of the FoXO transcription factor (Figure 4) [58,59]. The expression of these genes has already been associated with increased lifespan in *Drosophila* spp. [60]. PCA analysis associates peptide TRTX with the gene *sod2*, which is part of the InR/TOR pathway and may translate a cellular response to oxidative stress in the case of this treatment. On the other hand, SWD expression of the *diap1* gene was significantly higher for TRTX-treated flies. This gene encodes for *Drosophila* IAP1 (Diap1), a key anti-apoptotic enzyme that controls cell fate by hindering the activation of the initiator caspase Dronc, a caspase-9 homolog [61]. Increased transcription of the Diap1-expression gene indicates that peptide TRTX prompted the blockage of the apoptosis pathway, inhibiting the action of DrICE and Dcp-1, two downstream effector caspases (Figure 4).

Treatment with Hv1c may have triggered the expression of the heat shock protein (HSP) *hsc70-4*, according to PCA (Figure 4), which may be associated with some tolerance to this treatment. In fact, a study led by Gupta et al. [62] associated different susceptibilities to insecticides according to the expression or not of the *hsp70* gene. In the absence of *hsp70* expression, *D. melanogaster* accessory glands showed tissue damage in response to two insecticides, suggesting that the expression of this HSP is involved in the cellular response to the xenobiotics. Moreover, modulated expression of HSPs has also been associated with a longer lifespan in insects [60,63].

As previously suggested, other important barriers may be hindering peptide metabolization through the oral route. In fact, after ingestion, to reach the central nervous system where SVPs are believed to perform their main action, peptides must cross multiple physiological barriers, namely the insect cuticle, consisting of an apolar lipid matrix lining the fore and hindgut, and the peritrophic membrane covering the midgut of insect, both representing a major impairment to the delivery into the hemolymph [45,64]. Additionally, the presence of different pH values and proteases in the gut and physiological fluids may pose a threat to in vivo stability of peptides. Finally, the blood–brain barrier (BBB) adds an extra layer of difficulty in the access of the central nervous system [22]. Nonetheless, numerous proteins, such as bovine serum albumin (BSA) and immunoglobulins, have already shown an ability to traverse from the gut to the hemolymph in insects belonging to the *D. suzukii* order [64]. Moreover, Fitches et al. [16] showed a significant increase in the insecticidal activity of peptide Hv1a when fused with the snowdrop lectin *Galanthus nivalis* agglutinin (GNA), in comparison to Hv1a and GNA applied alone. When orally administered to *Mamestra brassicae* larvae, this molecular complex resulted in around 80% larval mortality within 10 days. The group also evaluated the presence of Hv1a/GNA in the insect hemolymph and nerve chords, as GNA binds to glycoproteins in the gut and flows through the membrane [65], indicating that the conjugate is able to reach the target [16].

Adding SVPs to regular fly food could impact food palatability and hence food intake, which could cause a change in survivorship. To date, there are no studies exploring the possibility that SVPs may or may not promote a change in some organoleptic characteristics of the food; our own initial attempts at measuring food intake gave inconsistent results (data not shown). However, a previous study showed that the commercial product Spear P (Vestaron Corporation), containing the active ingredient SVP GS-omega/kappa-Hxtx-Hv1a, promoted a higher mortality of *D. suzukii* flies when orally ingested and a decrease in *D. suzukii* infestation when the peptide was sprayed into a blueberry production [21]. In both application cases, no change in the palatability of the fruit was indicated.

Altogether, the activation of these stress response pathways may have hampered peptide activity, resulting in lower mortality for the concentrations assayed. On the other hand, the fore-mentioned biological barriers due to oral ingestion may have affected peptide bioavailability and metabolization, thus lowering its activity. These observations prompt new research questions that should be addressed to maximize SVPs’ potential as insecticides in general and against *D. suzukii* in particular.

## 5. Conclusions

The present work focused on assessing the insecticidal activity of two SVPs in *D. suzukii*. This study is the first to explore the biological response of SWD to peptides at a molecular level. Following the in vivo data and gene expression analysis, these peptides, especially Hv1c, may be considered for incorporation into an SWD control tool to be applied alone or in the scope of an integrated pest management approach. Future work should address the study of controlled oral delivery of peptides to tackle possible barriers to peptides’ metabolization. A deeper evaluation of the biological response of *D. suzukii* flies is necessary through total RNA sequencing (RNA-seq) to identify other possible peptide-triggered responses. Altogether, these data will allow a detailed understanding of challenges to the oral activity of SVPs, thus enabling the design of improved approaches to answer the SWD threat.

## Figures and Tables

**Figure 1 insects-14-00533-f001:**
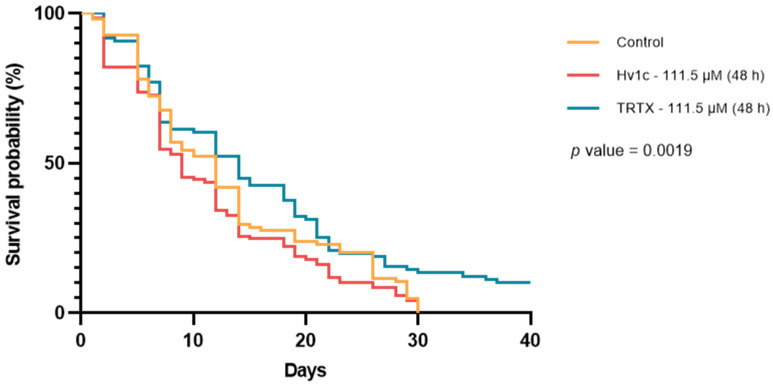
Kaplan–Meier survival curves of *Drosophila suzukii* flies orally exposed to 111.5 µM of Hv1c (red line) and TRTX (blue line) for 48 h, and respective control group (orange line). Y-axis indicates the probability of survival of flies (in percentage) over time. Only the results of the first 40 days after exposure to treatments are represented.

**Figure 2 insects-14-00533-f002:**
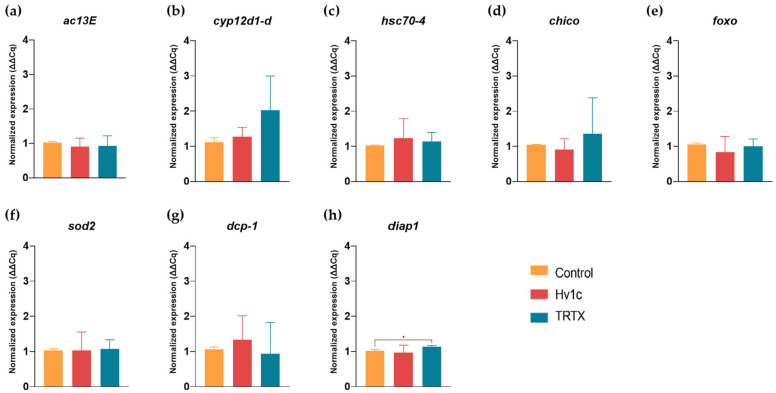
Normalized expression levels of different genes in response to oral ingestion of Hv1c and TRTX (111.5 µM for 48 h) by *Drosophila suzukii* flies. Transcripts associated with cytochrome P450 pathway: (**a**) adenylyl cyclase13E (*ac13E*); (**b**) cytochrome P450 12d1 distal (*cyp12d1-d*). Transcript associated with HSPs expression: (**c**) heat shock protein 70 cognate 4 (*hsc70-4*). Transcripts associated with insulin-like receptor/TOR pathway: (**d**) substrate of the product of InR (*chico*); (**e**) forkhead box, sub-group O (*foxo*); (**f**) superoxide dismutase 2 (*sod2*). Transcripts associated with apoptotic stimuli: (**g**) death caspase-1 (*dcp-1*); (**h**) death-associated inhibitor of apoptosis 1 (*diap1*). Vertical bars: mean values with standard deviation (*n* = 3); * represents statistically significant differences (*p* < 0.05).

**Figure 3 insects-14-00533-f003:**
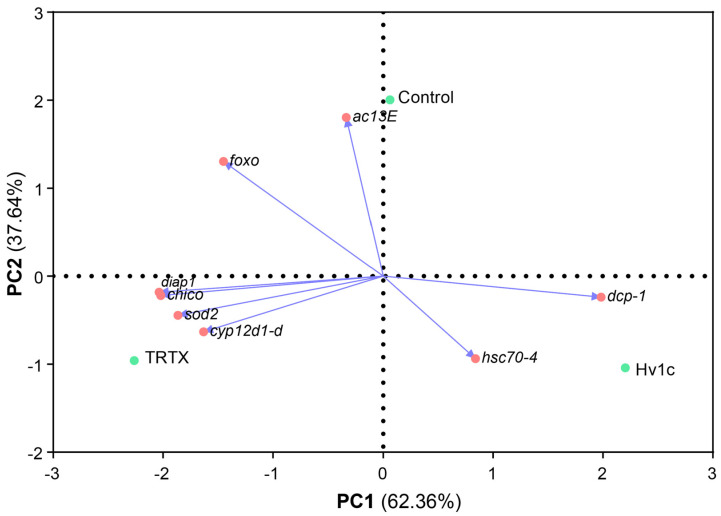
Principal component analysis biplot of the results from gene expression in *Drosophila suzukii* after treatment with Hv1c and TRTX at 111.5 µM (48 h of exposure). Abbreviations: *ac13E*—adenylyl cyclase 13E; *foxo*—forkhead box, sub-group O; *sod2*—superoxide dismutase 2; *cyp12d1-d*—cytochrome P450 12d1 distal; *dcp-1*—death caspase-1; *diap1*—death-associated inhibitor of apoptosis 1; *chico*—substrate of the product of InR; *hsc70-4*—heat shock protein 70 cognate 4.

**Figure 4 insects-14-00533-f004:**
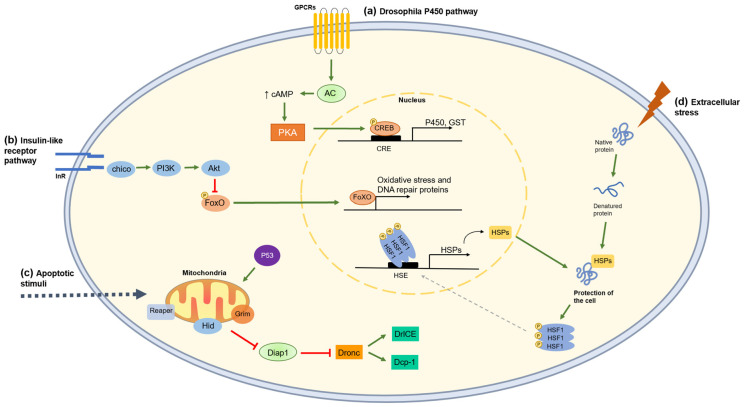
Proposed scheme for the gene expression pathways activated as responses to xenobiotics in *D. suzukii*: (**a**) when xenobiotics reach the cells, GPCRs may activate the adenylyl cyclase (AC) pathway, increasing the cAMP level in the cell, which ultimately binds to the cAMP-response element binding protein (CREB) that in turn links to the cAMP-response element (CRE) in the nucleus, promoting the expression of genes encoding for cytochromes P450 and glutathione-S-transferases (GSTs); (**b**) the insulin-like receptor/TOR signaling pathway (the last not represented) interprets growth factors, oxygenation state, and nutrition levels, fostering appropriate changes in cell growth and proliferation, survival, and fecundity; (**c**) under adverse conditions (e.g., DNA damage), apoptosis may be initiated by the regulation of pro-apoptotic proteins, i.e., Reaper, Hid, and Grim, these three forming a complex that inhibits the apoptosis inhibitor Diap1 (encoded by gene *diap1*). This inhibition unlocks the initiation caspase Dronc, consequently leading to the activation of effector caspases DrICE and Dcp-1; (**d**) heat shock proteins (HSPs) are expressed to protect cell components under stressful conditions.

**Table 1 insects-14-00533-t001:** *Drosophila suzukii* primers used for real-time quantitative PCR analysis.

Gene Symbol	Gene ID ^a^	Primer Sequences (5′ to 3′) ^b^	Reference
*tbp*	DS10_00003466	F: CCACGGTGAATCTGTGCT	[34]
		R: GGAGTCGTCCTCGCTCTT	
*argK*	DS10_00003811	F: CTACCACAACGATGCCAAGA	
		R: AAGGTCAGGAAGCCGAGA	
*ef1a48D*	DS10_00002426	F: TGGGCAAGGAAAAGATTCAC	[33]
		R: CGGCCTTCAACTTATCCAAA	
*hsc70-4*	DS10_00009978	F: TGCTGATCCAGGTGTACGAG	
		R: CGTTGGTGATGGTGATCTTG	
*foxo*	DS10_00012524	F: CTCCCTGAACACGTACAGCA	
		R: CTTCGACATTGCACTCCAGA	
*sod2*	DS10_00003278	F: TGGGAGCACGCCTACTATCT	This study
		R: GTCGTCCCAGTTAGCGATGT	
*chico*	DS10_00000455	F: TTATTTGGCGATGTCAACCA	
		R: GCTCTGGAAAGTCGAAATGC	
*ac13E*	DS10_00006210	F: TCACCTCGTTGAGCATGAAG	
		R: GGATGGATAATGCCACGTTC	
*cyp12d1-d*	DS10_00002643	F: GACGGTCTGGATTCGATTGT	
		R: TCGTCTTGTGAAGCAACCAG	
*dcp-1*	DS10_0002956	F: ACACGCCGACTTTCTGTTCT	
		R: CCAGGAGCCGTTGTTAATGT	
*diap1*	DS10_00011088	F: CGGGCATGTTCTACACACAC	
		R: GGGCAGATCCTCCTGCTC	

^a^ In SpottedWingFlyBase (http://spottedwingflybase.org/ (accessed on 1 June 2022); ^b^ F and R refer to forward and reverse primers, respectively.

## Data Availability

All the data are contained within the article and Appendix A.

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
