# Peer review of "Towards a Sustainable Management of the Spotted-Wing Drosophila: Disclosing the Effects of Two Spider Venom Peptides on Drosophila suzukii"

_insects, 2023, doi:10.3390/insects14060533_

Round 1
Reviewer 1 Report
As a reviewer of this journal article, I have reviewed the manuscript with ID insects-2398983 titled "Towards a sustainable management of the spotted-wing drosophila: disclosing the effects of two spider venom peptides on Drosophila suzukii". I recommend this manuscript for publication with minor revisions.
The manuscript provides an important insight into the efficacy of two spider venom peptides (SVPs) as a potential alternative to current control methods for the spotted-wing drosophila (Drosophila suzukii). The study demonstrates that one of the SVPs, µ-theraphotoxin-Hhn2b (TRTX), can enhance fly longevity compared to the control group, and also shows that detoxification and stress-related mechanisms are triggered in D. suzukii flies in response to these treatments.
However, there are a few points that need to be addressed before the manuscript can be published. Firstly, the figures (Figure 1-3) are disorganized and require more detailed explanations. Secondly, there are some language issues that need to be addressed for clarity and coherence.
Therefore, I suggest the authors revise the figures and provide more detailed explanations. Additionally, they should proofread the manuscript for language issues to improve the clarity and coherence of the text.
Overall, the study provides valuable information for the development of new insecticidal compounds for the management of the spotted-wing drosophila.
As a reviewer of this journal article, I have reviewed the manuscript with ID insects-2398983 titled "Towards a sustainable management of the spotted-wing drosophila: disclosing the effects of two spider venom peptides on Drosophila suzukii". I recommend this manuscript for publication with minor revisions.
The manuscript provides an important insight into the efficacy of two spider venom peptides (SVPs) as a potential alternative to current control methods for the spotted-wing drosophila (Drosophila suzukii). The study demonstrates that one of the SVPs, µ-theraphotoxin-Hhn2b (TRTX), can enhance fly longevity compared to the control group, and also shows that detoxification and stress-related mechanisms are triggered in D. suzukii flies in response to these treatments.
However, there are a few points that need to be addressed before the manuscript can be published. Firstly, the figures (Figure 1-3) are disorganized and require more detailed explanations. Secondly, there are some language issues that need to be addressed for clarity and coherence.
Therefore, I suggest the authors revise the figures and provide more detailed explanations. Additionally, they should proofread the manuscript for language issues to improve the clarity and coherence of the text.
Overall, the study provides valuable information for the development of new insecticidal compounds for the management of the spotted-wing drosophila.
Author Response
Reviewer 1:
As a reviewer of this journal article, I have reviewed the manuscript with ID insects-2398983 titled "Towards a sustainable management of the spotted-wing drosophila: disclosing the effects of two spider venom peptides on Drosophila suzukii". I recommend this manuscript for publication with minor revisions.
The manuscript provides an important insight into the efficacy of two spider venom peptides (SVPs) as a potential alternative to current control methods for the spotted-wing drosophila (Drosophila suzukii). The study demonstrates that one of the SVPs, µ-theraphotoxin-Hhn2b (TRTX), can enhance fly longevity compared to the control group, and also shows that detoxification and stress-related mechanisms are triggered in D. suzukii flies in response to these treatments.
However, there are a few points that need to be addressed before the manuscript can be published. Firstly, the figures (Figure 1-3) are disorganized and require more detailed explanations. Secondly, there are some language issues that need to be addressed for clarity and coherence.
Therefore, I suggest the authors revise the figures and provide more detailed explanations. Additionally, they should proofread the manuscript for language issues to improve the clarity and coherence of the text.
Overall, the study provides valuable information for the development of new insecticidal compounds for the management of the spotted-wing drosophila.
R: The authors thank the reviewer for the positive comment on their work. The manuscript was extensively checked to improve language clarity and text coherence. Figures 1-3 were described in more detail.
Reviewer 2 Report
In this manuscript Regalado et.al., tested two orally delivered spider venom peptides (SVP) as a potential compound to control Drosophila suzukii, one of the most invasive pest species in the world. The authors carried out survivorship experiments and looked at the expression of genes involved in stress resistance and the xenobiotic detoxification pathway. Overall, I find this manuscript interesting but lacks details that are necessary for repeating some of the experiments, specifically the survivorship experiment. I request authors to revise the manuscript and provide details that are missing.
My comments are attached below. I will be happy to review the revised manuscript.
- Which sex was used for the lifespan and survivorship experiments?
- What is the rationale for using only 30 minutes of starvation? Flies can survive longer starvation. Please specify how the starvation experiment was carried out, including whether flies were kept in vials with agar or moist tissue paper for starvation.
- What is the group size for the survivorship experiment? How frequently were flies transferred to fresh food vials during the survivorship experiments? Were flies kept in a same sex or mixed sex (male-female) environment during the survivorship experiment.
- Three biological replicates were used for the survivorship. Please specify how many flies were used for the experiments.
- Typo # 205 “with 800 μL of Massachusetts, USA”? Bracket is missing.
- Feeding differs in male and female flies. Please specify what sex was used for oral administration of the SVP. Please specify whether authors used any food dye to measure the quantity of Hv1c and TRTX that the fly ate at the onset of survivorship.
- Adding spider venom peptide (SVP) to regular fly food could change the palatability of the food, which could cause a change in survivorship. Please discuss these issues in the discussion.
Author Response
Reviewer 2:
In this manuscript Regalado et.al., tested two orally delivered spider venom peptides (SVP) as a potential compound to control Drosophila suzukii, one of the most invasive pest species in the world. The authors carried out survivorship experiments and looked at the expression of genes involved in stress resistance and the xenobiotic detoxification pathway. Overall, I find this manuscript interesting but lacks details that are necessary for repeating some of the experiments, specifically the survivorship experiment. I request authors to revise the manuscript and provide details that are missing.
My comments are attached below. I will be happy to review the revised manuscript.
R: The authors thank the reviewer for the generally positive assessment of their work. Overall, the comments and suggestions were accepted, and the authors answer bellow every point raised by the reviewer.
- Which sex was used for the lifespan and survivorship experiments?
R: For the lifespan and survivorship experiments, we used both male and female flies in equal numbers, as detailed and clarified in section 2.1. Biological material. Please see lines 114-115: “In every survival experiment, 2-5 days old flies were used (5 males and 5 females per experimental vial).”
- What is the rationale for using only 30 minutes of starvation? Flies can survive longer starvation. Please specify how the starvation experiment was carried out, including whether flies were kept in vials with agar or moist tissue paper for starvation.
R: The authors acknowledge the point raised by the reviewer. In fact, flies can survive longer starvation periods, and there are also studies employing starvation of up to 24 hours, but the authors intended not to force extra stress on the flies. Moreover, we wanted to mimic the conditions flies may face in nature, where they usually don’t have scarce food during the production season.
For starvation, flies were sorted in pools (5 males and 5 females each) and kept for 30 min in empty vials, without any source of hydration. This information was added to the manuscript for clarification. Please see lines 193-194: “D. suzukii flies were starved for 30 min in empty vials (…)”.
- What is the group size for the survivorship experiment? How frequently were flies transferred to fresh food vials during the survivorship experiments? Were flies kept in a same sex or mixed sex (male-female) environment during the survivorship experiment.
R: The survival experiment was performed with 3 biological replicates comprising 3 technical replicates of 10 flies each (5 males and 5 females). The information on the number of male and female flies per vial is also detailed in section 2.1. Biological material, as previously answered in point number 1.
Regarding the transfer to fresh food vials, flies were housed in new vials as pupae appeared, which means they were transferred every 7-10 days. This information is described in section 2.5. Survival assays. Please see lines 196 and 197: “Adults were removed and housed in new vials as pupae appeared.”
Regarding the last point raised, there was no separation between sexes in all experiments, so female and male flies were kept together, as previously described (Sario et al., 2018, doi: 10.1016/j.mrgentox.2018.05.001).
- Three biological replicates were used for the survivorship. Please specify how many flies were used for the experiments.
R: The authors thank the reviewer for this remark. For the survival experiments, at least three biological replicates were used. Each biological replicate corresponded to three technical replicates, each containing 5 male and 5 female flies, leading to a total of 90 flies per treatment/control group. We clarified this information in lines 198 and 199: “For each condition, at least three biological replicates were made (n = 3, for a total of 90 flies per condition).”
- Typo # 205 “with 800 μL of Massachusetts, USA”? Bracket is missing.
R: The authors thank the reviewer for drawing our attention to this error. The typo was corrected, the bracket was added, and the sentence was rephrased with missing information, as follows: “Briefly, samples were homogenized with 800 µL of NZYol with a 2.8 mm ceramic bead in a Fisherbrand™️ Bead Mill 24 (Thermo Fisher Scientific, Massachusetts, USA) (…)”.
- Feeding differs in male and female flies. Please specify what sex was used for oral administration of the SVP. Please specify whether authors used any food dye to measure the quantity of Hv1c and TRTX that the fly ate at the onset of survivorship.
R: The authors recognize the point brought up by the reviewer. It is known that male and female flies differ in feeding rates. However, in this study we opted to use both sexes for the oral administration assays, which is described in lines 114 and 115: “In every survival experiment, 2-5 days old flies were used (5 males and 5 females per experimental vial).”
Regarding the second point, this is a pertinent question from the reviewer. In the first stages of this study, the authors explored the possibility of measuring the SVP intake by the flies, by adding a blue dye to the food. The protocol was followed based on Wong et al., 2009 (doi: 10.1371/journal.pone.0006063). Although this information could be valuable to the interpretation of the results and to validate the feeding method, we had no consistent results, and the protocol does not take into account the flies’ excretion rates, so the results were unreliable. However, our study followed with the application of the SVPs embedded in a filter paper (after protocol optimization), so this makes it unfeasible to quantify the food and therefore the SVP intake.
- Adding spider venom peptide (SVP) to regular fly food could change the palatability of the food, which could cause a change in survivorship. Please discuss these issues in the discussion.
R: The authors appreciate this comment from the reviewer, which is a very interesting point of discussion. There are not, however, to date studies exploring the possibility that SVPs may or may not promote a change in some organoleptic characteristics of the food.
Nevertheless, a previous study (Fanning et al., 2018, doi: 10.1007/s10340-018-1016-7) showed that the commercial product Spear P (produced by Vestaron Corporation), in which the active ingredient is the SVP GS-omega/kappa-Hxtx-Hv1a, promoted a higher mortality of D. suzukii flies when orally ingested. Moreover, the same study demonstrated a decrease in D. suzukii infestation when the peptide was sprayed into a blueberry production, and in both cases, it was not observed any alteration that indicates a change in the palatability of the fruit.
With this, the authors decided not to include any information regarding this in the discussion, due to the reasons mentioned above.